# MultiDAN: Unsupervised, Multistage, Multisource and Multitarget Domain Adaptation for Semantic Segmentation of Remote Sensing Images

## ABSTRACT

Unsupervised domain adaptation (UDA) has been a crucial way for cross-domain semantic segmentation of remote sensing images and reached apparent advents. However, most existing efforts focus on single source single target domain adaptation, which don't explicitly consider the serious domain shift between multiple source and target domains in real applications, especially inter-domain shift between various target domains and intra-domain shift within each target domain. In this paper, to address simultaneous inter-domain shift and intra-domain shift for multiple target domains, we propose a novel unsupervised, multistage, multisource and multitarget domain adaptation network (MultiDAN), which involves multisource and multitarget domain adaptation (MSMTDA), entropy-based clustering (EC) and multistage domain adaptation (MDA). Specifically, MSMTDA learns feature-level multiple adversarial strategies to alleviate complex domain shift between multiple target and source domains. Then, EC clusters the various target domains into multiple subdomains based on entropy of target predictions of MSMTDA. Besides, we propose a new pseudo label update strategy (PLUS) to dynamically produce more accurate pseudo labels for MDA. Finally, MDA aligns the clean subdomains, including pseudo labels generated by PLUS, with other noisy subdomains in the output space via the proposed multistage adaptation algorithm (MAA). The extensive experiments on the benchmark remote sensing datasets highlight the superiority of our MultiDAN against recent state-of-the-art UDA methods.

## CCS CONCEPTS

• **Computing methodologies** → **Computer vision problems**.

## KEYWORDS

Multisource and Multitarget domain Adaptation (MSMTDA), Semantic Segmentation, Remote Sensing Images

## 1 INTRODUCTION

Recently, as the continuous increasing of remotely sensed images, semantic segmentation [5, 6] has shown impressive progress in

**Unpublished working draft. Not for distribution.**

remote sensing image interpretation applications, for example, climate change assessment, urban management, land cover monitoring, etc. Though these semantic segmentation models can achieve satisfactory performances [9] in a fully-supervised manner, they suffer from ubiquitous domain shift problems [28, 39, 41] in practical uses. That is to say, when the classifiers trained on training data (source domain) are directly employed to segment remote sensing images (target domain) drawn from different data distribution, their model performance will apparently drop.

To address the aforesaid issues, unsupervised domain adaptation (UDA) has been widely adopt to replace manual annotating for unlabeled images, and shown remarkable advances for cross-domain semantic segmentation in remote sensing community [14, 28, 41]. In summary, the majority of UDA methods focus on aligning source domain and target domain in the feature space [27, 45], output space [35, 36] and image space [2, 4]. On this basis, some self-supervised [22, 33] and stage-wise [3, 26, 43] adaptation methods are proposed to to further eliminate the serious UDA problems. Although the existing UDA approaches have shown significant progress for semantic segmentation of remote sensing images, most methods focus on single source single target domain adaptation (SDA) settings [3, 28, 41]. Such SDA settings limit the performance of existing UDA methods in real-world applications. Since in remote sensing community, each remote sensing image can be viewed as a single domain [34] due to various imaging modes. Thus the training data and test data usually involves multiple source domains and multiple target domains.

To fully exploit the multiple domain knowledge, some multi-source single target domain adaptation (MSDA) methods [19, 29, 44, 49] explore the abundant information from multiple source domains to adapt the classifier on a single target domain. Moreover, a few single source multitarget domain adaptation (MTDA) methods [24, 32, 50, 51] transfer the classifier trained on a single source domain to multiple target domains and exploit complementary knowledge among various target domains. Furthermore, some multisource and multitarget domain adaptation (MSMTDA) methods [34, 37] learn sufficient and complementary information from multisource and multitarget domains simultaneously, which can better eliminate the serious domain shift between multiple source and target domains against the MSDA and MTDA methods. However, in the MSMTDA scenarios, besides the domain shift across multisource and multitarget domains, there still exists severe multiple domain shift problem. As depicted in Figure 1, multiple domain shift problem involves inter-domain shift across various target domains along with simultaneous intra-domain shift within each target domain, which hasn't been well addressed by the existing MSMTDA methods. Specifically, the inter-domain shift between different target domains is caused by differences in imaging process. The intra-domain shift within each single domain usually

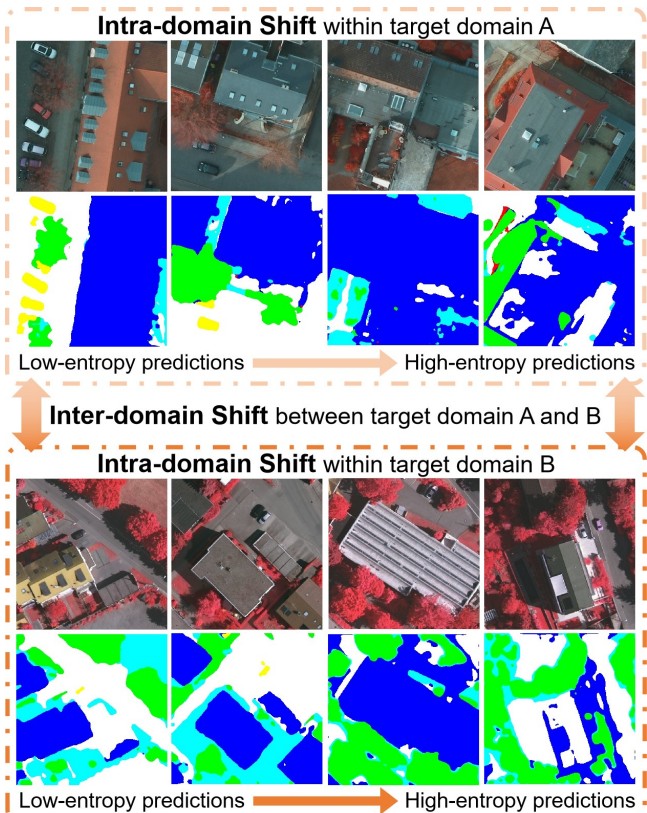

**Figure 1: Sample of multiple domain shift problem involving simultaneous inter-domain shift between various target domains and intra-domain shift within each target domain in the MSMTDA problem. For each target domain, the top row depicts sampled target images acquired from the same domain, with diverse roof colors and building styles. The bottom row demonstrates the corresponding predictions of Deeplabv3+ adapted from the same source domains.**

results from intra-class variation, illumination, diverse scene distributions and other factors. From Figure 1, we can see that the adapted target classifier can output many accurate and low-entropy predictions on both target domains after MSMTDA. However, there still exist some high-entropy and terrible predicted results for both target domains. This indicates the simultaneous inter-domain shift and intra-domain shift will significantly cripple the segmentation performance of MSMTDA models.

In this paper, we propose a novel unsupervised, multistage, multisource and multitarget domain adaptation network named MultiDAN, which includes multisource and multitarget domain adaptation (MSMTDA), entropy-based clustering (EC) and multistage domain adaptation (MDA). Concretely, the MSMTDA module firstly learns feature-level multiple adversarial strategies to mitigate the source-target domain shift between multiple target and source domains as well as target-target domain shift across various target domains, with adaptive weighting strategy (AWS) to reduce the manual efforts of hyperparameter tuning. Second, built upon the

confidence level of target predictions generated by MSMTDA module, EC clusters the various target domains into more fine-sorted subdomains for MDA module to solve the multiple domain shift problem. Then, to generate more confident pseudo labels for MDA module, we propose a novel pseudo label update strategy (PLUS), which dynamically update pseudo labels with high-confident and low-entropy predictions during every self-training stage. At last, to reduce the inter-domain shift between subdomains, MDA module adopts output-level multiple adversarial strategies and AWS to align the clean subdomains, including pseudo labels produced by PLUS, with other noisy subdomains by the proposed multistage adaptation algorithm (MAA).

In conclusion, our main contributions are as follows:

1) We reveal a crucial discovery that existing multisource and multitarget domain adaptation methods neglect the multiple domain shift problem involving simultaneous intra-domain shift and inter-domain shift within multiple target domains. Thus we propose a novel multistage, multisource and multitarget unsupervised domain adaptation network called MultiDAN for remotely sensed semantic segmentation.

2) We propose a novel multistage adaptation algorithm (MAA) to the multiple domain shift for MSMTDA, while the exiting self-supervised learning methods handle the multiple target domains as a whole and neglect the simultaneous inter-domain shift and intra-domain shift.

3) We propose a new pseudo label update strategy (PLUS) to dynamically update high-confidence or low-entropy pseudo labels for unlabeled target domains, while the exiting pseudo label strategies don't dynamically adopt the more confident pseudo labels during each training iteration.

## 2 RELATED WORK

### 2.1 Single Source Single Target Domain Adaptation

The aim of single source single target domain adaptation (SDA) is to adapt a transferable classifier to align the domain shift between one unlabeled target domain and one annotated source domain. Recently, SDA has been widely utilized for cross-domain semantic segmentation of natural images [1, 20, 27, 35, 36, 45] and remote sensing images [3, 4, 14, 28, 41]. For instance, Benjdira *et al.* [2] firstly trained a CycleGAN [52] to align the target and source domains in the image space. Then they applied the transformed target-like images to train the target segmentation model. On this basis, Li *et al.* [22, 33] further adopted high-confident predicted results of target classifier as pseudo labels, so as to optimize the target classifier according to self-supervised learning. Besides, some stage-wise UDA methods [3, 26, 43] proposed to further deal with the intra-domain shift within target domain, after mitigating the inter-domain shift between target domain and source domain. Although such recent SDA approaches have significantly bridged the domain gap between single source domain and single target domain, these methods don't explicitly consider multiple target and source domains, which would be more complex and difficult to solve in real applications.

## 2.2 Multisource Domain Adaptation and Multitarget Domain Adaptation

Multisource domain adaptation (MSDA) focuses on alleviating the distributional discrepancy between multiple source domains and single target domain [8, 12, 19, 29, 40, 46, 47], while Multitarget domain adaptation (MTDA) aims to alleviate the domain shift between single source domain and multiple target domains [10, 13, 42, 51]. For MSDA problems, Lu *et al.* [11, 23] utilized multiple incomplete source domains to form the categories of target domain. Zhao *et al.* [46, 47] transformed multisource images to target-like domains and eliminate the pixel-level distribution gap between multisource images. For MTDA issues, Gholami *et al.* [10] proposed an information theoretic model and explored the common intrinsic space for the single source and multitarget data. Isobe *et al.* [13] extended [25] to semantic segmentation task and added a pixel-wise regularization to enhance the cross-domain segmentation performance. However, the MSDA methods [16, 21, 30, 48, 49] don't explicitly take into account complex distribution discrepancy across various target domains while the MTDA methods [25, 31, 32] can't fully exploiting the advantageous knowledge from multiple source domains. The significant misalignment across simultaneous multitarget and multisource will apparently weaken the model performance of MSDA and MTDA methods. Thus, the MSDA and MTDA methods still lead to sub-optimal solutions when multisource and multitarget domains are available.

## 2.3 Multisource and Multitarget Domain Adaptation

Multisource and multitarget domain adaptation (MSMTDA) is proposed to eliminate the domain shift between multiple target domains and multiple source domains. However, only very few studies address the MSMTDA problem [34, 37, 38], as it is much more difficult and complex than SDA, MSDA and MTDA problems. Tasar *et al.* [34] adopted image translation models to conduct image-level alignment between multiple target domains and multiple source domains of satellite images. Then they trained the segmentation model on the diversified target-like images, which were randomly transferred from source images to one of the target domains, to make the segmentation model more robust to various target domains. Wang *et al.* [37] proposed to utilize numerous adversarial strategies to align each pair of source and target domains as well as each pair of target domains, so as to learn the common features of multiple target and source domains. Wu *et al.* [38] conducted hierarchical structure-level, domain-level and class-level alignments between multiple source and target domains. Although the beforementioned MSMTDA models can better handle the complex domain shift issues between multisource and multitarget data, they don't explicitly consider the intra-domain shift within every target domain alone with the inter-domain shift between various target domains, which has been validated to deteriorate the model performances. As a result, there are still some terrible predicted results for multitarget images after MSMTDA.

## 3 METHOD

We formalize the problem of MSMTDA for semantic segmentation, where $M$ labeled source domains $\{\mathcal{D}_S^1, \mathcal{D}_S^2, ..., \mathcal{D}_S^M\}$ and $N$ unlabeled target domains $\{\mathcal{D}_T^1, \mathcal{D}_T^2, ..., \mathcal{D}_T^N\}$ are available. Specifically, each source domain $\mathcal{D}_S^i$ ($i \in \{1, 2, ..., M\}$) includes source images $x_s^i \in \mathbb{R}^{H \times W \times 3}$ with $C$-category pixel-level annotations $y_s^i \in (1, C)^{H \times W \times C}$, while every target domain $\mathcal{D}_T^p$ ($p \in \{1, 2, ..., N\}$) involves target images $x_t^p \in \mathbb{R}^{H \times W \times 3}$ with no annotations.

As shown in Figure 2, the proposed MultiDAN consists of three parts. The first part is MSMTDA module adopting feature-level multiple adversarial strategies to reduce domain shift across multiple source and target domains. Secondly, EC computes the mean entropy of target predictions generated by MSMTDA module, and then uses the entropy to cluster the diversified target images obtained from various target domains into multiple subdomains for MDA module. Thirdly, MDA module applies output-level multiple adversarial strategies to align the clean subdomains, included pseudo labels generated by our PLUS, with other noisy subdomains via the proposed MAA.

## 3.1 Multisource and Multitarget Domain Adaptation

The MSMTDA module follows the main spirit of multiple adversarial frameworks [27, 37], which cope with the multisource and multitarget domain shift problem in an effective and simple way. Besides, an adaptive weighting strategy (AWS) [17] is applied to enhance the performance of segmentation model without extra manual efforts of hyperparameter tuning.

*3.1.1 Multiple Adversarial Domain Adaptation.* The MSMTDA module trains the source-target segmentation model $F_{ST}$ to learn the common features across multiple source and multiple target domains. Because there is no annotations for the multiple target domains, the source-target segmentation model $F_{ST}$ is optimized on the multiple source domains in a fully-supervised manner. To be specific, for every source domain $\mathcal{D}_S^i$ ($i \in \{1, 2, ..., M\}$) included images $x_s^i$ along with labels $y_s^i$, we train the source-target classifier $F_{ST}$ through minimizing the popular cross entropy loss with class-balanced factors [7]:

$$\mathcal{L}_{seg}^{s_i}(F_{ST}) = - \sum_c \alpha_c (y_s^i)^c \log(F_{ST}(x_s^i)^c) \quad (1)$$

where $\alpha_c$ is a class-balanced factor for every class $c \in C$. $\alpha_c$ denotes the inverse class frequency of effective number of class $c$ [7]. And $\alpha_c$ can be computed as: $\alpha_c = \frac{1-\beta}{1-\beta^{n_c}}$, where $n_c$ represents the pixel number of class $c$. $\beta$ is constant and set to 0.999. Moreover, $\alpha_c$ is normalized as $\alpha_c = \frac{\alpha_c}{\sum_{c=1}^C \alpha_c}$ to make $\sum_{c=1}^C \alpha_c = 1$, which enforces all the $\alpha_c$ ($c \in C$) within a close scope.

In order to alleviate the source-target domain shift between multiple target and source domains, we adopt feature-level multiple adversarial strategies [27, 37] to learn the shared intrinsic feature space of multiple source and target domains. Specifically, to fully consider the context dependencies across multiple target domains and multiple source domains, we firstly pair every source domain with every target domain and obtain $M \times N$ pairs ($\{\mathcal{D}_S^1, \mathcal{D}_T^1\}$,..., $\{\mathcal{D}_S^M, \mathcal{D}_T^N\}$). For each pair of source-target domains, we align source domain with target domain in the feature space through adversarial learning [45]. We apply the source-target discriminator $D_{s_i t_p}$ to differentiate the features $U_s^i$ of $x_s^i$ from features $U_t^p$ of $x_t^p$. At the

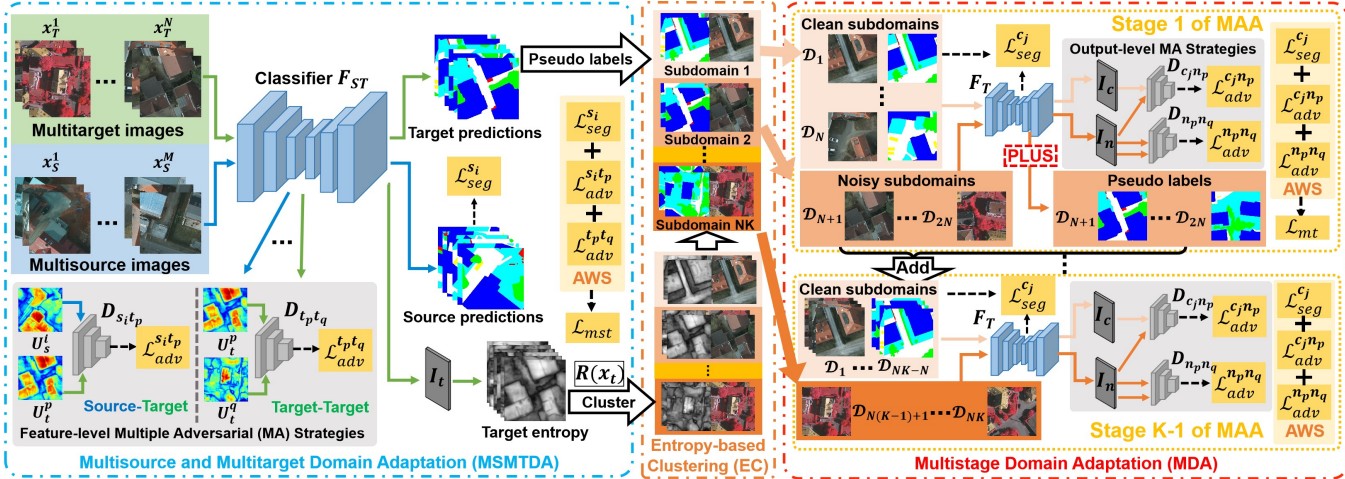

Figure 2: Overall training procedure of the proposed MultiDAN. The proposed MultiDAN includes multisource and multitarget domain adaptation (MSMTDA), entropy-based clustering (EC), and multistage domain adaptation (MDA). First, MSMTDA module trains the source-target segmentor $F_{ST}$ to reduce the domain shift between the multisource and multitarget domains based on feature-level multiple adversarial (MA) strategies. Second, EC clusters all the target domains into multiple subdomains based on ranking the mean entropy of target predictions of $F_{ST}$ generated by MSMTDA. Third, given the multiple subdomains, MDA module aligns the clean subdomains, which involves pseudo labels produced by the proposed *Pseudo Label Update Strategy (PLUS)*, with noisy subdomains via the proposed *Multistage Adaptation Algorithm (MAA)*. Moreover, for both MSMTDA and MDA modules, adaptive weighting strategy (AWS) is adopted to automatically learn the weights between various loss functions of optimization objectives $\lambda_{mst}$ and $\lambda_{mt}$. At test phase, the trained target segmentor $F_T$ can be directly applied to classify the target images without extra operations.

same time, the source-target classifier $F_{ST}$ is learned to extract domain-invariant features and confuse $D_{s_i t_p}$. Such source-target adversarial learning loss $\mathcal{L}_{adv}^{s_i t_p}$ for optimizing $F_{ST}$ and $D_{s_i t_p}$ on source-target pair $\{\mathcal{D}_S^i, \mathcal{D}_T^p\}$ ($i \in \{1, 2, ..., M\}$ and $p \in \{1, 2, ..., N\}$) can be formulated as

$$\mathcal{L}_{adv}^{s_i t_p}(F_{ST}, D_{s_i t_p}) = -(\log(1 - D_{s_i t_p}(U_s^i)) + \log(D_{s_i t_p}(U_t^p))) \quad (2)$$

where $U_s^i$ and $U_t^p$ are the features of $x_s^i$ and $x_t^p$.

Second, we pair every target domain with another target domain in a similar way and acquire $\frac{N \times (N-1)}{2}$ pairs ($\{\mathcal{D}_T^1, \mathcal{D}_T^2\}$,..., $\{\mathcal{D}_T^{N-1}, \mathcal{D}_T^N\}$). For each pair of target-target domains, we align the two different target domains in the feature space through adversarial learning [45]. The target-target discriminator $D_{t_p t_q}$ is adopted to distinguish the features $U_t^p$ of $x_t^p$ from features $U_t^q$ of $x_t^q$, while the classifier $F_{ST}$ is trained to fool $D_{s_i t_p}$. Such target-target adversarial learning loss $\mathcal{L}_{adv}^{t_p t_q}$ for optimizing $F_{ST}$ and $D_{t_p t_q}$ on target-target pair $\{\mathcal{D}_T^p, \mathcal{D}_T^q\}$ ($p, q \in \{1, 2, ..., N\}$ and $p \neq q$) can be expressed as

$$\mathcal{L}_{adv}^{t_p t_q}(F_{ST}, D_{t_p t_q}) = -(\log(1 - D_{t_p t_q}(U_t^p)) + \log(D_{t_p t_q}(U_t^q))) \quad (3)$$

where $U_t^p$ and $U_t^q$ are the features of $x_t^p$ and $x_t^q$.

It is notable that there should have been $M \times N$ source-target discriminators and $\frac{N \times (N-1)}{2}$ target-target discriminators, but we utilize two multitask discriminators $\{D_{s_i t_p}, D_{t_p t_q}\}$ (see Figure 3) instead, where discriminator $D_{s_i t_p}$ is used for different pairs of source domains and target domains, $D_{t_p t_q}$ is applied for different pairs of target domains. In detail, the multitask discriminator includes four

shared convolutional layers and one specific convolutional layers for different source-target pairs (or target-target pairs) as depicted in Figure 3. In this way, when the number of source and target domains increases, we only need to add specific convolutional layers, instead of adding discriminators.

*3.1.2 Adaptive Weighting Strategy.* To reduce time consumption and manual efforts for tuning hyperparameters as well as obtaining the optimal weights between every loss function, we adopt an adaptive weighting method [17] in our final optimization objective. It could adaptively learn the weights between the segmentation loss $\mathcal{L}_{seg}^{s_i}$, source-target adversarial learning loss $\mathcal{L}_{adv}^{s_i t_p}$ and target-target adversarial learning loss $\mathcal{L}_{adv}^{t_p t_g}$. The final adaptively weighted optimization objective of MSMTDA module can be expressed as

$$\mathcal{L}_{mst} = \sum_{i=1}^{M} \frac{1}{2\theta_{s_i}^2} \mathcal{L}_{seg}^{s_i} + \log(1 + \theta_{s_i}^2)$$
$$+ \sum_{i=1}^{M} \sum_{p=1}^{N} \frac{1}{2\theta_{m_{ip}}^2} \mathcal{L}_{adv}^{s_i t_p} + \log(1 + \theta_{m_{ip}}^2) \quad (4)$$
$$+ \sum_{p=1}^{N} \sum_{q=1, p \neq q}^{N} \frac{1}{2\theta_{t_{pq}}^2} \mathcal{L}_{adv}^{t_p t_q} + \log(1 + \theta_{t_{pq}}^2)$$

Where $i \in \{1, 2, ..., M\}$ and $p, q \in \{1, 2, ..., N\}$ ($p \neq q$). Parameters $\theta = \{\theta_{s_i}, \theta_{m_{ip}}, \theta_{t_{pq}}\}$ are learnable and adaptive during the training process.

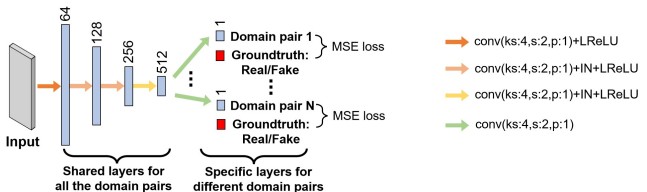

**Figure 3: Structure of multitask discriminators.** *Conv, p, s and ks* **stand for convolution layer, padding, stride and kernel size respectively.** *MSE, IN* **and** *LReLU* **represent Mean Squared Error, Instance Normalization and Leaky Rectified Linear Unit respectively. The number above the blue rectangles indicates the channel number in each activation.**

## 3.2 Entropy-based Clustering

To solve the inter-domain shift between various target domains and intra-domain shift within each target domain, we utilize entropy-based clustering (EC) to cluster the diversified multitarget domains into multiple subdomains in accordance with the confidence level of predictions generated by MSMTDA module. In this way, we can tackle the multiple inter-domain shift and intra-domain shift problems of multitarget domains as multiple inter-domain shift problem between different subdomains. Because of no target annotations, we utilize mean entropy of softmax probability $P_t$ ($P_t = F_{ST}(x_t)$) to assess the confidence level of target predictions [36]. The entropy $I_t$ can be computed as

$$I_t^{(h, w)} = - \sum_c P_t^{(h, w, c)} \log P_t^{(h, w, c)} \quad (5)$$

where $h \in H$ and $w \in W$. $H \times W$ denote the image size of target predictions.

Then, we rank and split all target images $x_t$ via calculating the mean entropy grade $R(x_t)$ of corresponding target predictions $P_t$. The mean entropy grade $R(x_t)$ can be defined as

$$R(x_t) = \frac{1}{HW} \sum_{h, w} I_t^{(h, w)} \quad (6)$$

On this basis, we propose to separate and rank $N$ target domains into multiple ($K \times N$) subdomains $\{\mathcal{D}_1, \mathcal{D}_2, ..., \mathcal{D}_{NK}\}$ and proposed a novel MAA to alleviate the complex multiple domain shift problem. Note that $K$ is a hyperparameter that controls the subdomain number for each target domain and stage number for MAA.

## 3.3 Multistage Domain Adaptation

To further reduce the domain shift between multiple subdomains, we propose a novel multistage domain adaptation (MDA) module. In general, we align the output space of clean subdomains, including pseudo labels produced by our PLUS, with that of noisy subdomains via the proposed MAA. And MDA module is also trained with AWS.

*3.3.1 Pseudo Label Update Strategy.* Because of lack of target annotations, it isn't practicable to directly adapt the target classifier on the multiple target subdomains. Some UDA methods [3, 26, 43] proposed to utilize the target predictions of the classifier as pseudo labels for self-supervised learning. However, there may still be some incorrect pixels in the clean predictions during the self-training

procedure, even if these clean predictions are relatively low-entropy and clean on the whole. To obtain the optimal pseudo labels during the iterative training process, we propose a novel pseudo label update strategy (PLUS) to produce more accurate pseudo labels as much as possible. Concretely, for each self-training stage, we apply the newly optimized target classifier $F_T$ to output predictions $P_t$ and compute entropy $I_t$ for $x_t$. Besides, we employ two-dimensional ($H \times W$) matrix $\mathcal{P}$ and $\mathcal{I}$, which have the same size as the predictions $P_t$ and entropy $I_t$, to store the maximum probability value and minimum entropy value of every pixel within target image $x_t$. Then for each pixel $x_t^{(h, w)}$ of $x_t$, we update pseudo labels by assigning the class label, which has a larger probability value $P_t^{(h, w, c)}$ than the corresponding value $\mu^{(h, w)} \in \mathcal{P}$ or has a smaller entropy value $I_t^{(h, w)}$ than the corresponding value $v^{(h, w)} \in \mathcal{I}$. At the same times, we update matrix $\mathcal{P}$ and $\mathcal{I}$ with the larger probability value or the smaller entropy value for each pixel. We formulate the proposed PLUS as follows:

$$\hat{y}_t^{(h, w)} = \begin{cases} c, \text{ if } P_t^{(h, w, c)} > \mu^{(h, w)} \text{ or } I_t^{(h, w)} < v^{(h, w)} \\ \text{and } \underset{\tilde{c}}{\operatorname{argmax}} P_t^{(h, w, \tilde{c})} = c \\ 0, \text{ otherwise} \end{cases} \quad (7)$$

where $\mu^{(h, w)} \in \mathcal{P}$ and $v^{(h, w)} \in \mathcal{I}$ are the probability threshold and entropy threshold for pixel $x_t^{(h, w)}$ of $x_t$ respectively.

The update strategy for matrix $\mathcal{P}$ and $\mathcal{I}$ can be expressed in the following equation:

$$\tilde{\mu}^{(h, w)} = \operatorname{MAX}\{P_t^{(h, w, c)}, \mu^{(h, w)}\}. \quad (8)$$

$$\tilde{v}^{(h, w)} = \operatorname{MIN}\{I_t^{(h, w)}, v^{(h, w)}\}. \quad (9)$$

where $\mu^{(h, w)} \in \mathcal{P}$ and $v^{(h, w)} \in \mathcal{I}$ are the existing values in matrix $\mathcal{P}$ and $\mathcal{I}$. $\tilde{\mu}^{(h, w)}$ and $\tilde{v}^{(h, w)}$ are the new values to be updated in matrix $\mathcal{P}$ and $\mathcal{I}$. The initial value of $\mu^{(h, w)}$ and $v^{(h, w)}$ are set to $\mu_0$ and $v_0$. MAX and MIN are functions which pick up the maximum value and minimum value between different values respectively.

*3.3.2 Multistage Adaptation Algorithm.* To address the severe domain shift between multiple subdomains, we propose a novel multistage adaptation algorithm (MAA), which utilizes multiple clean subdomains along with pseudo labels generated by PLUS to adapt other multiple noisy subdomains in a multistage manner as described in Algorithm 1.

Specifically, in every self-training stage, we applied the proposed PLUS to generate and update pseudo labels for subdomains $\{\mathcal{D}_1, ..., \mathcal{D}_{Nk}\}$. Second, we cluster and update subdomains $\{\mathcal{D}_1, ..., \mathcal{D}_{Nk}\}$ into $N$ clean subdomains $\{\mathcal{D}_{clean}^1, ..., \mathcal{D}_{clean}^N\}$ according to mean entropy grade $R(x_t)$. Then, we treat $N$ clean subdomains along with pseudo labels as the labeled multisource domains, and deal with $N$ noisy subdomains $\{\mathcal{D}_{Nk+1}, ..., \mathcal{D}_{N(k+1)}\}$ as unlabeled multitarget domains. Similar to MSMTDA module, we utilize a target segmentation module $F_T$ and two multitask discriminators $\{D_{c_j n_p}, D_{n_p n_q}\}$ to narrow the clean-noisy domain shift between multiple clean (multisource) subdomains and multiple noisy (multitarget) subdomains.

Firstly, we train the target segmentation model $F_T$ to extract the shared features across multiple clean subdomains and multiple noisy subdomains. Because of lack of annotations for the multiple noisy subdomains, the target segmentation model $F_T$ will be trained

---

**Algorithm 1:** Multistage Adaptation Algorithm

---

**Input**:

    The target subdomains $\{\mathcal{D}_1, \mathcal{D}_2, ..., \mathcal{D}_{NK}\}$

    The clean subdomains $\mathcal{D}_{clean} = \emptyset$

    The initial network $F_{ST}, D_{c_j n_p}, D_{n_p n_q}$

**Output**:

    The trained network $F_T^{(K-1)}, D_{c_j n_p}^{(K-1)}, D_{n_p n_q}^{(K-1)}$

**1** $F_T^{(0)} \leftarrow F_{ST}, D_{c_j n_p}^{(0)} \leftarrow D_{c_j n_p}, D_{n_p n_q}^{(0)} \leftarrow D_{n_p n_q}$

**2 for** $k = 1$ to $K - 1$ **do**

**3**      input images $\{x_1 \in \mathcal{D}_1, ..., x_{Nk} \in \mathcal{D}_{Nk}\}$ into $F_T^{(k-1)}$ and update their pseudo labels $\{\hat{y}_1, ..., \hat{y}_{Nk}\}$ with Equations (7), (8) and (9)

**4**      update $\{(x_1, \hat{y}_1) \in \mathcal{D}_1, ..., (x_{Nk}, \hat{y}_{Nk}) \in \mathcal{D}_{Nk}\}$ into $\mathcal{D}_{clean}$

**5**      cluster all $(x_{clean}, \hat{y}_{clean}) \in \mathcal{D}_{clean}$ into $N$ clean subdomains $\{\mathcal{D}_{clean}^1, ..., \mathcal{D}_{clean}^N\}$ based on $R(x_t)$ in Equation (6)

**6**      utilize $N$ clean subdomains $\{\mathcal{D}_{clean}^1, ..., \mathcal{D}_{clean}^N\}$ and $N$ noisy subdomains $\{x_{Nk+1} \in \mathcal{D}_{Nk+1}, ..., x_{N(k+1)} \in \mathcal{D}_{N(k+1)}\}$ to train $F_T^{(k)} \leftarrow F_T^{(k-1)}, D_{c_j n_p}^{(k)} \leftarrow D_{c_j n_p}^{(k-1)}, D_{n_p n_q}^{(k)} \leftarrow D_{n_p n_q}^{(k-1)}$ with Equations (10), (11), (12), (13)

**7 end**

**8** return $F_T^{(K-1)}, D_{c_j n_p}^{(K-1)}, D_{n_p n_q}^{(K-1)}$

---

on the multiple clean subdomains with pseudo labels in a self-supervised manner. For every clean subdomain $\mathcal{D}_j$ $(j \in \{1, ..., N\})$ included images $x_{clean}^j$ along with pseudo labels $\hat{y}_{clean}^j$, we adapt the target classifier $F_T$ via minimizing the class-balanced cross entropy loss [7]:

$$\mathcal{L}_{seg}^{c_j}(F_T) = -\sum_c \alpha_c (\hat{y}_{clean}^j)^c \log(F_T(x_{clean}^j)^c) \quad (10)$$

where $\alpha_c$ is a class-balanced factor as described earlier.

Secondly, to eliminate the inter-domain shift between multiple clean subdomains (source) and multiple noisy subdomains (target), we utilize output-level multiple adversarial strategies [27, 37] to learn the common intrinsic space of multiple subdomains. In detail, we firstly pair every clean subdomain with every noisy subdomain and acquire $N \times N$ pairs. For each pair of clean-noisy subdomains, we align clean subdomain with noisy subdomain in the output space by adversarial learning [36]. The clean-noisy adversarial learning loss $\mathcal{L}_{adv}^{c_j n_p}$ for optimizing classifier $F_T$ and clean-noisy discriminator $D_{c_j n_p}$ can be expressed as

$$\mathcal{L}_{adv}^{c_j n_p}(F_T, D_{c_j n_p}) = -(\log(1 - D_{c_j n_p}(I_c^j)) + \log(D_{c_j n_p}(I_n^p))) \quad (11)$$

where $I_c^j$ and $I_n^p$ are the entropy of clean subdomain $j \in \{1, ..., N\}$ and noisy subdomain $p \in \{Nk + 1, ..., N(k + 1)\}$. And entropy map $I$ can be calculated bu Equation (5).

Thirdly, we pair every noisy subdomain (target) with another noisy subdomain (target) in a similar way and obtain $\frac{N \times (N-1)}{2}$ pairs. For each pair of noisy-noisy subdomains, we align the different subdomains in the output space via adversarial learning [36].

The noisy-noisy adversarial learning loss $\mathcal{L}_{adv}^{n_p n_q}$ for optimizing classifier $F_T$ and noisy-noisy discriminator $D_{n_p n_q}$ $(p \neq q)$ can be written as

$$\mathcal{L}_{adv}^{n_p n_q}(F_T, D_{n_p n_q}) = -(\log(1 - D_{n_p n_q}(I_n^p)) + \log(D_{n_p n_q}(I_n^q))) \quad (12)$$

where $I_n^p$ and $I_n^q$ are the entropy of noisy subdomain $p$ and $q$ respectively $(p, q \in \{Nk + 1, ..., N(k + 1)\})$.

Similar to the adaptively weighted MSMTDA module, we use adaptive weighting method [17] in our final optimization objective of MDA module, which can adaptively learn the weights between the target segmentation loss $\mathcal{L}_{seg}^{c_j}$, clean-noisy adversarial learning loss $\mathcal{L}_{adv}^{c_j n_p}$ and noisy-noisy adversarial learning loss $\mathcal{L}_{adv}^{n_p n_q}$. The final adaptively weighted optimization objective of the proposed MDA module can be summarized as

$$\mathcal{L}_{mt} = \sum_{j=1}^N \frac{1}{2\sigma_{c_j}^2} \mathcal{L}_{seg}^{c_j} + \log(1 + \sigma_{c_j}^2)$$

$$+ \sum_{j=1}^N \sum_{p=1}^N \frac{1}{2\sigma_{m_{jp}}^2} \mathcal{L}_{adv}^{c_j n_p} + \log(1 + \sigma_{m_{jp}}^2) \quad (13)$$

$$+ \sum_{p=1}^N \sum_{q=1, p \neq q}^N \frac{1}{2\sigma_{n_{pq}}^2} \mathcal{L}_{adv}^{n_p n_q} + \log(1 + \sigma_{n_{pq}}^2)$$

Where $j \in \{1, ..., N\}$ and $p, q \in \{Nk + 1, ..., N(k + 1)\}$ $(p \neq q)$. Parameters $\sigma = \{\sigma_{c_j}, \sigma_{m_{jp}}, \sigma_{n_{pq}}\}$ are learnable and adaptive during the training process.

## 4 EXPERIMENTS

### 4.1 Experimental Setup

*4.1.1 Dataset.* We validate the proposed MultiDAN for MSMTDA problem on the public aerial image segmentation (AIS) datasets [15] collected form Tokyo, Berlin, Paris, Chicago, Potsdam and Zurich cities. Potsdam, Chicago, Paris and Zurich datasets include 24, 457, 625 and 364 annotated aerial images with near 3000 × 3000 resolution, respectively. Berlin and Tokyo datasets involve 200 and 1 annotated aerial images with about 2500 × 2500 resolution, respectively. For the semantic annotations, blue, white and red colors correspond to road, background and building respectively.

*4.1.2 Implementation Details.* We apply Deeplabv3+ [6] as semantic segmentation model $F$. The multitask discriminators $D$ utilize five convolutional layers as illustrated in Figure 3. During the training phase, MSMTDA and MDA modules of the proposed MultiDAN can be separately trained in sequence. Firstly, the MSMTDA module is trained for 100 epochs with loss function $\mathcal{L}_{mst}$ in Equation (4). The initial values of adaptive weighting parameter $\{\theta_{s_i}, \theta_{m_{ip}}, \theta_{t_{pq}}\}$ are uniformly set to 1, 0.05 and 0.02 respectively. In this stage, the segmentation model $F_{ST}$, discriminators $\{D_{s_i t_p}, D_{t_p t_q}\}$ and parameter $\theta$ are trained jointly via Adam optimizer [18] with $\beta_1 = 0.9$ and $\beta_2 = 0.999$. The batch size and learning rate are set to 12 and $10^{-4}$. Secondly, the trained segmentation model $F_{ST}$ is utilized to segment the multiple target domains. We calculate the entropy grade $R(x_t)$ (Equation (6)) of all the target predicted maps, and sort the target predictions with correspond images according to $R(x_t)$. Then the sorted target images are separated into multiple

**Table 1: Comparisons Between the Proposed MultiDAN and the Recent SOTA UDA Methods on the Adaptation From Multisource Zurich and Chicago Datasets to Multitarget Paris and Berlin Datasets.**

| Method | Paris | | | | | | | | Berlin | | | | | | | |
|---|---|---|---|---|---|---|---|---|---|---|---|---|---|---|---|---|
| | Background | | Building | | Road | | Avg | | Background | | Building | | Road | | Avg | |
| | F1 | IoU | F1 | IoU | F1 | IoU | mF1 | mIoU | F1 | IoU | F1 | IoU | F1 | IoU | mF1 | mIoU |
| Deeplabv3+ | 48.4 | 32.6 | 51.2 | 42.7 | 32.4 | 18.8 | 44.0 | 31.4 | 52.7 | 36.3 | 49.8 | 41.6 | 25.9 | 13.5 | 42.8 | 30.5 |
| IterDANet | 70.2 | 56.8 | 71.4 | 57.6 | 38.1 | 22.7 | 59.9 | 45.7 | 70.8 | 57.4 | 73.6 | 58.9 | 33.1 | 19.6 | 59.2 | 45.3 |
| CPSL | 70.7 | 57.2 | 71.8 | 57.4 | 38.9 | 23.2 | 60.5 | 45.9 | 70.5 | 57.1 | 72.4 | 58.0 | 31.8 | 18.2 | 58.2 | 44.4 |
| DRT | 71.2 | 57.8 | 72.7 | 58.6 | 43.6 | 28.3 | 62.5 | 48.2 | 72.2 | 58.8 | 72.9 | 59.7 | 41.3 | 26.7 | 62.1 | 48.4 |
| MADAN+ | 71.7 | 58.2 | 71.9 | 57.3 | 41.2 | 26.4 | 61.6 | 47.3 | 72.7 | 59.2 | 73.8 | 60.3 | 43.6 | 28.8 | 63.4 | 49.4 |
| CGCT | 71.8 | 58.0 | 72.8 | 58.8 | 44.2 | 28.9 | 62.9 | 48.6 | 73.2 | 59.4 | 73.5 | 60.1 | 42.0 | 27.3 | 62.9 | 48.9 |
| TSAN | 72.2 | 58.3 | 72.4 | 58.1 | 42.5 | 27.1 | 62.4 | 47.8 | 72.4 | 58.9 | 72.6 | 59.2 | 40.4 | 25.7 | 61.8 | 47.9 |
| DAugNet | 72.6 | 58.8 | 73.2 | 59.9 | 45.8 | 30.2 | 63.9 | 49.6 | 74.8 | 60.4 | 74.2 | 61.3 | 46.4 | 30.5 | 65.1 | 50.7 |
| MSTDA | 74.3 | 59.7 | 74.8 | 60.2 | 46.3 | 30.6 | 65.1 | 50.2 | 75.1 | 60.8 | 75.7 | 62.2 | 47.8 | 31.6 | 66.2 | 51.5 |
| MultiDAN | **75.1** | **60.3** | **76.9** | **61.5** | **48.8** | **32.8** | **66.9** | **51.5** | **76.4** | **61.3** | **77.6** | **63.7** | **51.3** | **34.5** | **68.4** | **53.2** |

**Table 2: Comparisons Between the Proposed MultiDAN and the Recent UDA Methods on the Adaptation From Multisource Paris, Berlin and Tokyo Datasets to Multitarget Zurich, Chicago and Potsdam Datasets.**

| Method | Potsdam | | | | | | Zurich | | | | | | Chicago | | | | | |
|---|---|---|---|---|---|---|---|---|---|---|---|---|---|---|---|---|---|---|
| | Building | | Road | | Avg | | Building | | Road | | Avg | | Building | | Road | | Avg | |
| | F1 | IoU | F1 | IoU | mF1 | mIoU | F1 | IoU | F1 | IoU | mF1 | mIoU | F1 | IoU | F1 | IoU | mF1 | mIoU |
| Deeplabv3+ | 51.8 | 44.2 | 28.7 | 16.2 | 45.7 | 33.5 | 46.7 | 39.8 | 27.8 | 15.8 | 42.6 | 30.9 | 51.1 | 42.4 | 28.6 | 16.7 | 42.7 | 31.6 |
| IterDANet | 76.4 | 61.5 | 32.2 | 20.7 | 60.3 | 46.9 | 67.6 | 54.2 | 33.4 | 21.2 | 56.9 | 43.9 | 67.2 | 53.9 | 38.5 | 25.1 | 58.5 | 45.2 |
| CPSL | 79.2 | 63.2 | 32.8 | 21.2 | 61.3 | 47.5 | 71.7 | 58.3 | 34.3 | 22.5 | 58.7 | 45.9 | 67.7 | 54.5 | 38.4 | 24.7 | 58.4 | 45.1 |
| DRT | 78.7 | 62.6 | 42.4 | 27.1 | 64.6 | 49.5 | 74.1 | 60.8 | 41.2 | 26.1 | 62.7 | 48.8 | 68.3 | 55.3 | 42.8 | 27.9 | 60.2 | 46.6 |
| MADAN+ | 77.4 | 62.1 | 43.9 | 28.5 | 64.9 | 50.0 | 73.5 | 59.9 | 41.5 | 26.4 | 62.4 | 48.3 | 68.5 | 55.6 | 43.5 | 28.6 | 60.0 | 46.5 |
| CGCT | 76.3 | 61.2 | 41.7 | 26.8 | 64.0 | 49.3 | 71.8 | 59.2 | 40.6 | 25.9 | 61.3 | 47.5 | 70.6 | 57.7 | 44.3 | 29.2 | 61.2 | 47.6 |
| TSAN | 76.7 | 61.8 | 41.3 | 26.2 | 63.9 | 49.2 | 71.2 | 58.7 | 41.1 | 26.2 | 61.5 | 48.0 | 70.8 | 58.1 | 43.8 | 28.5 | 61.3 | 47.7 |
| DAugNet | 75.8 | 60.5 | 44.6 | 29.4 | 64.3 | 49.5 | 73.3 | 59.4 | 46.5 | 30.6 | 64.8 | 50.2 | 69.4 | 56.3 | 45.3 | 29.8 | 61.4 | 47.6 |
| MSTDA | 78.6 | 62.8 | 45.9 | 30.1 | 66.4 | 51.0 | 76.4 | 62.1 | 46.1 | 30.3 | 65.8 | 51.2 | 70.6 | 57.8 | 45.7 | 30.5 | 62.2 | 48.7 |
| MultiDAN | **80.3** | **64.6** | **48.6** | **32.5** | **68.4** | **52.8** | **78.5** | **64.8** | **50.7** | **33.6** | **68.9** | **54.1** | **73.8** | **60.6** | **47.2** | **32.4** | **64.5** | **50.7** |

subdomains evenly by setting $K = 4$. Thirdly, the MDA module are leaned with Algorithm 1. Specifically, the initial values of probability threshold in Equation (8) and entropy threshold in Equation (9) are determined by pixel ratio $\rho_u = 75\%$ and $\rho_v = 65\%$. The initial values of adaptive weighting parameter $\{\sigma_{s_i}, \sigma_{m_{ip}}, \sigma_{t_{pq}}\}$ are uniformly set to 1, 0.05 and 0.02 respectively. In every self-training stage, the segmentation model $F_T$, discriminators $\{D_{c_j n_p}, D_{n_p n_q}\}$ and adaptive weighting strategy $\sigma$ are jointly trained for 100 epochs through Adam optimizer [18] with $\beta_1 = 0.9$ and $\beta_2 = 0.999$. The batch size and learning rate are set to 12 and $10^{-4}$. All experiments are conducted on two NVIDIA Tesla P40 GPUs.

## 4.2 Results and Discussions

We compare our MultiDAN with recently published state-of-the-art (SOTA) UDA models involving SDA methods [3, 20], MSDA methods [21, 46], MTDA methods [31, 51] and MSMTDA methods [34, 37]. When training the SDA, MSDA and MTDA methods on the multisource and multitarget domains, we follow the related works

[12, 13, 30–32, 46, 51] and integrate all source domains or target domains into one source domain or one target domain respectively.

Tables 1 and 2 show the segmentation results of the proposed MultiDAN and the SOTA UDA methods. The baseline Deeplabv3+ performs the worst on multisource and multitarget AIS datasets. All the UDA models lead to apparent performance improvements. We can find that MSDA methods and MTDA methods are generally better than the SDA models. At the same time, MSMTDA approaches surpass the MSDA methods and MTDA methods. These results show the effectiveness and superiority of MSMTDA methods, which address the severe domain shift between multiple source and target domains while the MSDA and MTDA methods ignore the useful knowledge across multitarget data or multisource data. Among all the MSMTDA models, the proposed MultiDAN yields the highest mIoU and mF1. This highlights our MultiDAN has more advantages in MSMTDA tasks compared with the SOTA UDA approaches. Figures 4 and 5 draw the visual segmentation results of the UDA models of Tables 1 and 2 respectively.

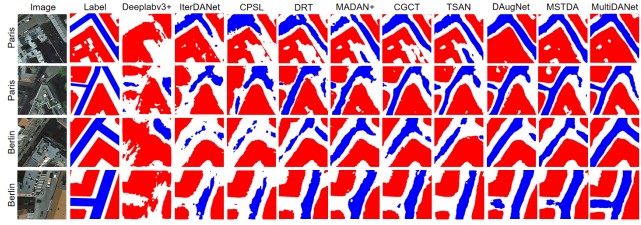

**Figure 4: Visual comparisons between the proposed Multi-DAN and the recent SOTA UDA methods on the adaptation from multisource Zurich and Chicago datasets to multitarget Paris and Berlin datasets.**

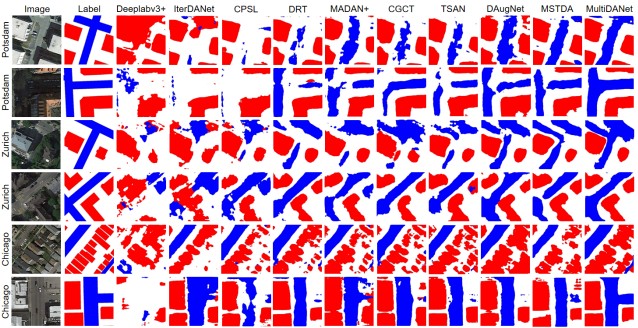

**Figure 5: Visual comparisons between the proposed Multi-DAN and the recent SOTA UDA methods on the adaptation from multisource Paris, Berlin and Tokyo datasets to multitarget Zurich, Chicago and Potsdam datasets.**

## 4.3 Ablation Study

*4.3.1 Components Analysis.* We apply Tokyo, Berlin and Paris datasets as source domain, while Potsdam, Zurich and Chicago datasets as target domain, and conduct components analysis of AWS, MAA, PLUS. MSTDA denotes the baseline MSMTDA module without AWS. As shown in Table 3, every component and combination of components can bring performance improvements over the baseline method, which verifies the effectiveness of each component of our MultiDAN in MSMTDA tasks.

**Table 3: Component Analysis of the Proposed MultiDAN.**

| Method | Components | Potsdam | Zurich | Chicago |
|---|---|---|---|---|
| Model-I | MSTDA | 48.3 | 47.9 | 46.4 |
| Model-II | MSTDA+AWS | 48.9 | 48.6 | 47.1 |
| Model-III | MSTDA+MAA | 50.8 | 51.7 | 49.2 |
| Model-IV | MSTDA+MAA+AWS | 51.5 | 52.6 | 49.6 |
| Model-V | MSTDA+MAA+AWS+PLUS | **52.8** | **54.1** | **50.7** |

*4.3.2 Influence of Stage Number K.* We probe the effect of stage number $K$ by training MultiDAN with various stage number $K$, and report the results in Table 4. From Table 4, we can see the segmentation performances of MultiDAN firstly improve and then

turn slightly worse with the continuous increasing of stage number $K$ (from 5 to 8).

**Table 4: Influence of Stage Number (Subdomain Number) $K$.**

| $K$ | 1 | 2 | 3 | 4 | 5 | 6 | 7 | 8 |
|---|---|---|---|---|---|---|---|---|
| Paris | 46.9 | 48.7 | 50.3 | **51.5** | 51.3 | 51.2 | 50.9 | 50.8 |
| Berlin | 48.3 | 50.6 | 52.5 | **53.2** | 52.8 | 52.6 | 52.3 | 51.9 |

*4.3.3 Comparing Different Pseudo Label Strategy.* We compare the proposed PLUS and the popular pseudo label strategy adopted in recent SOTA UDA frameworks. Table 5 gives the quantitative performance of directly utilizing target predictions without pseudo label strategy (No), softmax probability threshold (ST) [22], entropy threshold (ET) [33], combination of softmax probability threshold and entropy threshold (EST) [3] and our PLUS (Ours). As shown in Table 5, all the pseudo label approaches surpass directly utilizing target predictions without pseudo label strategy (No), proving the necessity of pseudo label strategies. Among all the pseudo label strategies, our PLUS achieves the best segmentation performance, which highlights the competitiveness of our PLUS.

**Table 5: Segmentation Performances (mIoU) of the Proposed MultiDAN With Various Pseudo Label Strategies.**

| Method | No | ST | ET | EST | Ours |
|---|---|---|---|---|---|
| Potsdam | 51.5 | 52.2 | 52.3 | 52.6 | **52.8** |
| Zurich | 52.6 | 53.2 | 53.6 | 53.7 | **54.1** |
| Chicago | 49.6 | 50.3 | 50.2 | 50.6 | **50.7** |

*4.3.4 Supplementary Materials.* In supplementary materials, we show the visual feature distributions of our MultiDAN and comparing methods, and visual segmentation predictions and entropy the proposed MultiDAN with various stage numbers (subdomain number) $K$. Then, we discuss the methods of determining the initial values of probability threshold $\mu$ in Equation (8) and entropy threshold $v$ in Equation (9), and probe the effect of initial values of $\mu$ and $v$. Besides, we validate the effectiveness of adaptive weighting strategy (AWS) and study the effect of different source and target domains.

## 5 CONCLUSION

This paper proposes a multistage, multisource and multitarget UDA network called MultiDAN to further solve serious multiple domain shift problem in practical applications of remote sensing images, consisting of simultaneous inter-domain shift between various target domains and intra-domain shift within each target domain. Extensive experiments on the open-source benchmark remote sensing data sets demonstrates the competitiveness and superiority of the proposed MultiDAN over the existing SOTA UDA models.

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
