# OpenReview forum: "MultiDAN: Unsupervised, Multistage, Multisource and Multitarget Domain Adaptation for Semantic Segmentation of Remote Sensing Images"
_acmmm.org/ACMMM/2024/Conference — MM2024 Poster_

### Official Review · Reviewer_c5Ac · 2024-05-22

**Rating:** 2
**Confidence:** 3

**Summary:**

This paper proposes to handle the Multisource and Multitarget Domain Adaptation for Semantic Segmentation of Remote Sensing Images by adopting the MultiDAN method. Specifically, the multistage adaptation algorithm (MAA) mainly focuses on the inter-domain shift and the intra-domain shift. The pseudo-label update strategy (PLUS) is utilized to dynamically update the high-confidence or low-entropy pseudo-labels for unlabeled target domains. Experimental evaluation shows the effectiveness of the proposed method.

**Strengths:**

*	The authors reveal a critical gap in existing MSMTDA methods, which often overlook the issue of multiple domain shifts, including both intra-domain and inter-domain shifts within multiple target domains.
*	Experimental analysis is thorough, where the previous methods not tailored for the new task are reproduced.

**Limitations:**

*	In Fig. 1, the authors illustrate the low-entropy and high-entropy predictions generated by the adopted target classifier. However, the comparison between low-entropy predictions and high-entropy predictions is not intuitive enough. Suitable statistical techniques can be conducted to enhance the clarity of Fig. 1.
*	In Sec. 3.2, the authors describe the entropy and the mean entropy grade in Eqs. 5 and 6, respectively. However, as the main topic of Sec. 3.2 is entropy-based clustering, the detailed steps of the clustering process can be further illustrated.
*	The authors claim that existing MSMTDA methods overlook the problems of inter- and intra-domain shifts. However, the inter- and intra-domain attention modules proposed in AMDA [1] already address these issues. The authors are suggested to further highlight the superiority of their approach in addressing inter- and intra-domain shifts compared to AMDA.
*	In Sec 4.2, the authors claim they compare their method with the MSMTDA method AMDA. However, the relevant experiment results are not illustrated on Tabs. 1 and 2.

[1] Yuxi Wang, Zhaoxiang Zhang, Wangli Hao, and Chunfeng Song. 2020. Attention guided multiple source and target domain adaptation. IEEE TIP, 30: 892-906.

**Suitability:**

2

---

### Official Review · Reviewer_swu1 · 2024-05-24

**Rating:** 3
**Confidence:** 3

**Summary:**

This paper proposes a multi-stage framework for the task of Semantic Segmentation of Remote Sensing Images. The proposed method involves mainly three stages including a network for target prediction, a clustering module, and multistage domain adaptation. The authors aim to solve the issues of both inter- and intra-domain shift. Experimental results show that the proposed method performs favorably against the SOTA methods on the evaluated datasets.

**Strengths:**

1. The motivation is well explained and the research context is clearly presented.
2. The experimental results of the method are good and the implementation details are described in detail.

**Limitations:**

1. While the authors emphasize the importance of addressing intra- and inter-domain shifts simultaneously, the paper fails to demonstrate what bad effects can these two issues cause, respectively. On the contrary, these two issues are dealt with as a single issue as “the problem of the mixture of multi-source images”. Moreover, the experiments did not verify how the inter-domain and intra-domain issues were solved.
2. Concerning the tuning of hyper-parameters, although the paper presents a method of learning the parameters in equation 4, more theoretical and experimental justifications are needed to show why optimizing equation 4 can lead to the optimal solution of the problem
involving the three loss functions.
3. The model complexity of the methods in table 1 and 2 should be compared, since we can infer from the paper that the proposed method may require a large number of model parameters and FLOPs.
4. It is unclear why the model performance continues to become worse as the model is trained with more stages when K exceeds 4. After all, the added stages are supposed to refine the results of previous stages.

**Suitability:**

2

---

### Official Review · Reviewer_6ish · 2024-05-25

**Rating:** 5
**Confidence:** 3

**Summary:**

This paper proposes a unsupervised, multistage, multisource and multitarget domain adaptation network (MultiDAN), which involves multisource and multitarget domain adaptation (MSMTDA), entropy-based clustering (EC) and multistage domain adaptation (MDA). Besides, a new pseudo label update strategy (PLUS) is proposed. The function of each module is clear and the whole target of the paper is well illustrated.

**Strengths:**

1. The paper is well designed and the paper writing is clear.
2. The experiments are adequate.
3. Each module is designed to address a problem, and it is reasonable to do so.

**Limitations:**

1. What about the selection of K affecting the computation complexity?
2. Can the PLUS be used on other methods?
3. How are the clean and noisy subdomains identified?

**Suitability:**

2

---

### Meta-Review · Area_Chair_AuJB · 2024-07-01

**Recommendation:** Accept (Poster)
**Confidence:** 4

**Metareview:**

The paper proposes a novel approach for the task of semantic segmentation for remote sensing data. Reviewers noted the paper's clear design and well-illustrated goals, with each module addressing specific problems effectively. While some concerns about computational complexity, and the identification of clean and noisy subdomains were raised, the authors' responses were mostly satisfactory. It is suggested that the authors carefully revise the paper in the camera-ready version according to reviewers' comments.